# Effect of drainage layers on water retention of potting media in containers

**Avery Rowe**[ID]*

Independent Scholar, Wales, United Kingdom

* avery@tradescantia.uk

**Data availability statement:** All relevant data are within the manuscript and its Supporting information files.

## Abstract

Excess water retention in the potting medium can be a significant problem for plants grown in containers due to the volume of saturated medium which forms above the drainage hole. Adding a layer of coarse material like gravel or sand at the bottom is a common practise among gardeners with the aim of improving drainage, but some researchers have argued that such layers will raise the saturated area and in fact increase water retention. Two different depths and four different materials of drainage layer were tested with three different potting media to determine the water retention in the container after saturating and draining freely. For loamless organic media, almost all types of drainage layer reduced overall water retention in the container compared to controls. For loam-based media, most drainage layers had no effect on the overall water retention. Two simple models were also used to estimate the water retention in the media alone, excluding the drainage layer itself. All drainage layers reduced water retention of loamless organic media, according to both models. There was disagreement between the two models applied to loam-based media, and further study is required to determine the most accurate. Both models showed that some drainage layers with smaller particle sizes reduced water retention in loam-based media, but disagreed on the effect of drainage layers with larger particle sizes. Overall, any drainage layer was likely to reduce water retention of any medium, and almost never increased it. Thicker drainage layers were more effective than thinner layers, with the most effective substrate depending on the potting media used. A 60 mm layer of coarse sand was the most universally-effective drainage layer with all potting media tested.

## Introduction

It is widely recognised that soil in a container behaves differently from soil in the field [1–3]. One of the most significant differences is the presence of a saturated volume of soil at the bottom of the container after irrigation and draining, known as a perched water table [4–6]. The depth of this saturated volume is constant for any specific soil, and unaffected by container size and shape. As a result, soil in a shallower container has a greater percentage water-holding capacity (WHC) than the same soil in a deeper container [1,7–9]. This effect has led to the concept of container capacity, a measure of the total water retained in a specific container filled with specific soil after saturation and drainage [4,9,10].

**Funding:** The author(s) received no specific funding for this work.

**Competing interests:** The authors have declared that no competing interests exist.

Adding a layer of rocks or gravel at the bottom of plant pots is a common practice among home gardeners, as well as some researchers [11], and is generally intended to reduce this perched water table and improve drainage. However in recent years the recommendation has been disputed, with arguments based on principles of water movement through soil suggesting that a drainage layer will in fact raise the perched water table and increase water retention overall [12,13]. Despite frequent discussion of this practice, there has been little formal research into the effect drainage layers have on the water-holding capacity of potting media in containers.

A number of areas of research may be applicable to this subject in part. One of the oldest relevant studies is the tests of water movement through soil interfaces conducted by Hsieh and Gardner [14]. This research demonstrated that when the wetting front encounters an interface between soils of different porosities, it will not advance until the soil behind is nearly saturated. Applying this to the interface between potting medium and drainage substrate in a container suggests that water will not move down into the drainage layer until the medium is almost saturated, and therefore the perched water table in the medium will be raised and the overall water retention increased.

More recent studies have demonstrated that a layer of coarse material acts as a barrier to water movement in field soils [15], and that more moisture is retained in root zone soil when a layer of gravel is added below it [16]. Khire et al. [17] also found that more water is stored above a capillary barrier layer when it consists of coarser material than when it consists of finer material. Smesrud and Selker [18] conducted theoretical analysis which predicted that the effectiveness of a capillary barrier is maximised when the contrast of particle sizes between the soil above and the substrate below is infinite - in other words, when the soil is suspended over a void of air. This would suggest that the perched water table created by soil over any other substrate would be less than that created by soil over air.

Another area of recent relevant research which can be applied to this subject is the construction of green roofs, which are roofs made from a layer of growing medium topped with living plants [19]. This shallow layer of medium is surrounded by an impermeable base and sides, with a small number of drains for excess water to escape through - physical conditions which are similar to a typical plant pot. In the construction of green roofs a drainage layer of coarse substrate below the growing medium is always considered essential [19,20]. She and Pang [21] found that water begins to drain out from the soil of the green roof after it has reached field capacity (container capacity), but before it is fully saturated. This result contrasted with predictions that water only escapes at full saturation, showing the need for further research to understand the movement of water through soil in containers with layered contents.

Ruter [22] found that when a fabric-bottomed container was placed on a column of sand to drain, the perched water table in the medium was completely eliminated. This suggests that water moved more easily from the medium to sand than from the medium into air, and therefore that adding a drainage layer of sand to the bottom of a container may aid drainage. However, the column of sand in this study was many times deeper than the container itself, so this may not predict the behaviour of a thin drainage layer. In addition, the fabric bottom of the container may have an effect on the water movement from the medium to the sand which would not occur if the two materials were in direct contact.

Many of the above studies were conducted on mineral field soils. In contrast, container growing often utilises loamless potting media based on organic materials like peat, coir, and bark, with mineral amendments like perlite and vermiculite [2,4,5,9,13,23,24]. These media typically have very different physical properties from loams, including higher total porosity and lower water-holding capacity [4,25]. As a result, the water movement through loamless

media in containers may not be accurately predicted by applying research conducted on field loams.

Further, researchers have demonstrated that the water movement and other behaviour of soil in containers is highly dependent on the geometry of the container [3], and can differ greatly from the behaviour of even the same soil in the field [2]. Thus, much of the research into the behaviour of water at soil interfaces cannot be applied directly to the situation of container growing because it was conducted in the field or in containers of very different sizes and shapes to those generally used for live plants.

A typical filled plant pot consists of an impermeable container with a small number of holes in the base, full of a single uniform potting medium. When the saturated medium is allowed to drain, the only interface the water passes through is the drainage hole where medium meets air. In contrast, a pot with a drainage layer is a more complex system. First the water passes through the interface between the medium and the drainage substrate, and then it passes through the interface between the drainage substrate and the air under the drainage hole. These multiple interactions make it more difficult to predict the final outcome using only theoretical analysis of individual soil interfaces.

In recent years, some growers have begun to stratify different growing media within containers, to deliberately adjust the vertical moisture profile [26]. Criscione et al. found that substrate stratification significantly changes the movement of water through containers [27,28], as well as the growth of plants in these containers [29]. These studies were all conducted using two layers of substrate each half the height of the container, and therefore the conclusions may not be applicable to the use of shallower drainage layers in the base of a container mostly filled with one medium.

This study aimed to determine the effect on water-holding capacity of drainage layers in containers, and compare the effect using different potting media, different drainage substrates, and different layer depths. The intention was to obtain results which could be applied by commercial nurseries and home gardeners in practice, so the conditions were chosen to closely replicate real growing containers. The study did not attempt to determine the effect of drainage layers on growing plants directly. Experiments and theoretical models were used to determine the water-holding capacity of the entire container, and of the medium alone, so that both results could be considered by readers when predicting the effects on plant growth.

## Materials and methods

Three different types of potting media were used, as follows: JI was a loam-based John Innes "No. 1" compost (Bathgate); CV was a mix of 70% coir (Coir Products Coco Peat) and 30% vermiculite (Westland Gro-Sure) by volume; CPB was a mix of 60% coir, 20% perlite (Vitax 2–6 mm), and 20% pine bark (Melcourt Propagating Bark 2–7 mm) by volume. The two loamless mixes were chosen to represent some of the most common loamless media used by growers according to Bunt [4]. Such media typically contain at least 50% peat or fine bark, with the remainder composed of mineral amendments (such as perlite, vermiculite, or sand), compost, or coarse bark. Coir was chosen as an acceptable alternative to peat in these mixes [30,31], due to the UK government's intention to reduce or eliminate horticultural use of peat.

Four different drainage substrates were used: gravel (Melcourt Horticultural Gravel 4–10 mm), leca (LECA Lightweight Aggregate 4–10 mm), grit (LECA Lightweight Aggregate 2–4 mm), and sand (Dehner Aquarienkies 1–2 mm). This range of substrates was chosen to examine the influence of particle size on drainage effects, as well as to compare the effect of porous substrates with non-porous ones.

Test pots were made from clear plastic containers with a single 10 mm hole cut in the centre of the base. The pots had a total capacity of 1,290 ml, and were an inverted truncated cone shape with height 175 mm, base diameter 86 mm, top diameter 118 mm, and dry weight of 53 g. The base had a slight domed area raised by 4 mm relative to the edges, common to nursery containers to allow water to escape when placed on a flat surface. For trials with sand, a piece of fibreglass mesh (16 x 18 mesh count, approximately 1 mm spacing) was placed over the drainage hole to prevent the sand escaping. The size and friction of particles of the other drainage substrates was sufficient to prevent them escaping from the drainage hole. The drainage hole was temporarily sealed with tape before filling.

For each potting medium, trials were conducted with a layer of each drainage substrate to depths of 30 mm and 60 mm (giving a total of 24 combinations). Control trials were also conducted without drainage substrate with the pot filled with media to the top, to 30 mm from the top, and to 60 mm from the top. In addition, trials were conducted of all the 30 mm and 60 mm drainage layers with no media. Ten replications were conducted for each condition, with the exception of two conditions (CV-leca-30 mm and sand-60 mm) which each had one invalid trial for a total of nine results.

For each of the three potting media, a baseline moisture level was defined according to the mass of a fixed volume of medium. Before each condition, the medium was air-dried until it reached this moisture level, confirmed by weighing a fixed volume of medium to ensure it had the correct baseline mass. All the containers were then filled with equal masses of medium (as recommended in the procedure Bilderback [32] gives for measuring soil porosity in the nursery) to within 5 g precision, using a scale with 1 g precision. The medium was added by hand, and no evidence of layering or spaces was observed through the clear containers. Each filled container was irrigated from above using a watering can with a rose until the water level reached the top of the pot (or to the specified height for the tests involving pots filled to below the top). Finally the drainage hole was unsealed and the container allowed to drain suspended on a grate and uncovered, as a growing pot would be. The containers were monitored until water was no longer visibly draining, which took between one minute and three hours, depending on the condition. No evidence of blockages or inconsistencies was observed during drainage. The pot was then weighed a final time (to 1g precision) with its wetted contents. The mass of the empty container was subtracted from the measurements to determine the initial dry mass ($m_i$) and drained mass ($m_d$) of the contents alone. See Fig 1 for diagrams of the experimental procedure.

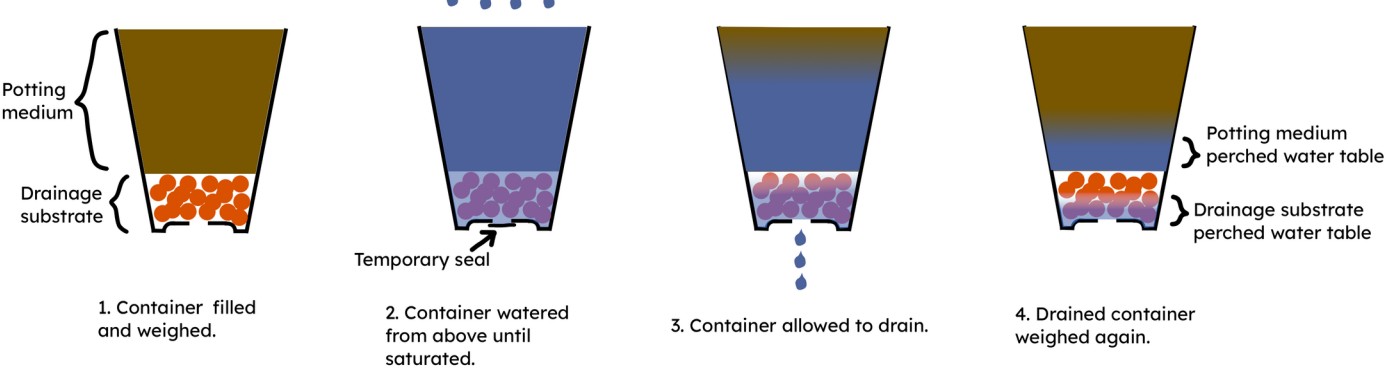

**Fig 1. Experimental procedure.**

The saturating and draining process caused all the potting media to compact. The compacted depth was measured from one representative sample for each medium with each depth of drainage substrate, and the final compacted volume calculated ($v$). None of the drainage substrate layers compacted measurably, so their final volume was taken as equal to the initial volume.

The water-holding capacity (WHC) was then calculated for each trial (see Eq 1). ANOVA post-hoc Tukey tests were used to determine the significance of differences between the means of different conditions (see S1 Appendix for ANOVA tables). Different potting media were not compared directly to each other, because their compacted volumes and initial moisture levels were not standardised for this study.

$$WHC = (m_d - m_i)/v \tag{1}$$

Two simple models were created to estimate the WHC of the medium itself in containers with drainage layers. In both models, the initial mass of the medium in each trial ($m_{i(med)}$) was calculated using the initial mass of the full container ($m_{i(full)}$) and the initial mass of the corresponding drainage layer ($m_{i(drn)}$) (see Eq 2).

$$m_{i(med)} = m_{i(full)} - m_{i(drn)} \tag{2}$$

Model A was intended to estimate the highest possible WHC for the medium itself. The model was based on the assumption that the drainage layer held the same amount of water after draining when in a container filled with medium as when in a container alone. Thus, the drained mass of the medium in each trial according to model A ($m_{d(A)}$) was calculated by subtracting the drained mass of the drainage layer ($m_{d(drn)}$) from that of the full container ($m_{d(full)}$)(see Eq 3).

$$m_{d(A)} = m_{d(full)} - m_{d(drn)} \tag{3}$$

Model B was intended to estimate the lowest possible WHC for the medium itself. The model was based on the assumption that the drainage layer remained fully saturated with water after draining when in a container filled with medium. Thus, the drained mass of the medium in each trial according to model B ($m_{d(B)}$) was calculated by subtracting the saturated mass of the drainage layer ($m_{s(drn)}$) from the drained mass of the full container (see Eq 4).

$$m_{d(B)} = m_{d(full)} - m_{s(drn)} \tag{4}$$

The combination of these two models provided a range of possible values for WHC of the medium alone in each condition, to use as a basis for comparison against the control condition using the same depth of medium. This enabled the water retained in the drainage layer itself to be eliminated as a factor, and the effect of the drainage layer on the medium above it to be considered in isolation.

## Results

### Coir-perlite-bark medium

The CPB mix showed the least compaction, with the effective volume of mix in a full pot decreasing from 1,290 ml to 1,133 ml. The control container filled to 30 mm from the top

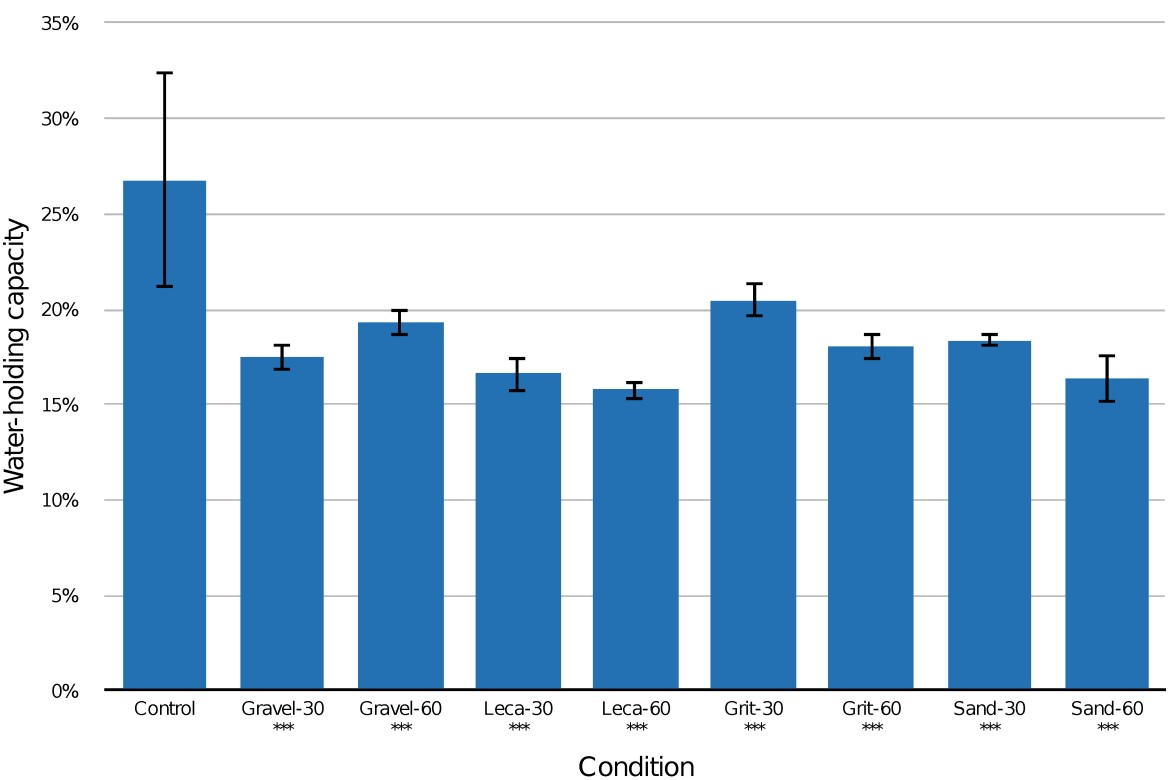

**Fig 2. Mean and standard deviation water-holding capacity (WHC) of the entire container of CPB medium with different drainage layer types.** Asterisks mark conditions which were significantly different from the control (*** $p < 0.001$).

compacted from 1,030 ml to 838 ml, and the container filled to 60 mm from the top compacted from 780 ml to 672 ml. The WHC of the controls increased as the depth of medium decreased.

The results of the CPB drainage layer trials are presented in Fig 2 by combined drainage layer type. Post-hoc Tukey comparisons are presented in Table 1 (by combined drainage layer type), Table 2 (by drainage substrate only), and Table 3 (by drainage layer depth only). Compared to the control, every container with a drainage layer had a lower overall WHC ($p < 0.001$). There were significant differences between many of the drainage layer conditions, with grit-30 mm condition having the highest WHC and leca-60 mm the lowest. Analysing only according to the drainage substrate, all had lower overall WHC than the control ($p < 0.001$). Leca had significantly lower WHC than gravel ($p < 0.05$) and grit ($p < 0.001$). Combining all substrates, both 30 mm and 60 mm drainage layer conditions had lower overall WHC than the control ($p < 0.001$) with no significant difference between them.

The modelled WHC of the medium itself according to drainage layer substrate is presented in Fig 3 (30 mm drainage layers) and Fig 4 (60 mm drainage layers), compared with the control containers with the same initial medium depth. With both drainage layer depths, both the modelled WHC values were lower than the control values for every condition.

## Coir-vermiculite medium

The CV mix showed intermediate compaction, with the effective volume of mix in a full pot decreasing from 1,290 ml to 1062 ml. The control container filled to 30 mm from the top

**Table 1. Post-hoc Tukey comparisons of water-holding capacity of the entire container of CPB medium with different drainage layer types.**

| Drainage layer type | Comparison | Mean Difference | SE | t | p |
|---|---|---|---|---|---|
| Gravel-60 mm | Grit-60 mm | 0.012 | 0.009 | 1.352 | 0.912 |
| | Leca-60 mm | 0.035 | 0.009 | 3.773 | 0.009** |
| | Sand-60 mm | 0.029 | 0.009 | 3.126 | 0.059 |
| | Gravel-30 mm | 0.017 | 0.009 | 1.883 | 0.627 |
| | Grit-30 mm | −0.012 | 0.009 | −1.360 | 0.909 |
| | Leca-30 mm | 0.027 | 0.009 | 2.911 | 0.101 |
| | Sand-30 mm | 0.008 | 0.009 | 0.904 | 0.992 |
| | Control | −0.076 | 0.009 | −8.258 | < .001*** |
| Grit-60 mm | Leca-60 mm | 0.022 | 0.009 | 2.422 | 0.287 |
| | Sand-60 mm | 0.016 | 0.009 | 1.774 | 0.699 |
| | Gravel-30 mm | 0.005 | 0.009 | 0.531 | 1.000 |
| | Grit-30 mm | −0.025 | 0.009 | −2.712 | 0.160 |
| | Leca-30 mm | 0.014 | 0.009 | 1.559 | 0.823 |
| | Sand-30 mm | −0.004 | 0.009 | −0.448 | 1.000 |
| | Control | −0.088 | 0.009 | −9.610 | < .001*** |
| Leca-60 mm | Sand-60 mm | −0.006 | 0.009 | −0.648 | 0.999 |
| | Gravel-30 mm | −0.017 | 0.009 | −1.891 | 0.622 |
| | Grit-30 mm | −0.047 | 0.009 | −5.134 | < .001*** |
| | Leca-30 mm | −0.008 | 0.009 | −0.863 | 0.994 |
| | Sand-30 mm | −0.026 | 0.009 | −2.870 | 0.112 |
| | Control | −0.110 | 0.009 | −12.032 | < .001*** |
| Sand-60 mm | Gravel-30 mm | −0.011 | 0.009 | −1.243 | 0.944 |
| | Grit-30 mm | −0.041 | 0.009 | −4.486 | < .001*** |
| | Leca-30 mm | −0.002 | 0.009 | −0.215 | 1.000 |
| | Sand-30 mm | −0.020 | 0.009 | −2.222 | 0.402 |
| | Control | −0.104 | 0.009 | −11.384 | < .001*** |
| Gravel-30 mm | Grit-30 mm | −0.030 | 0.009 | −3.243 | 0.043* |
| | Leca-30 mm | 0.009 | 0.009 | 1.028 | 0.982 |
| | Sand-30 mm | −0.009 | 0.009 | −0.979 | 0.987 |
| | Control | −0.093 | 0.009 | −10.141 | < .001*** |
| Grit-30 mm | Leca-30 mm | 0.039 | 0.009 | 4.271 | 0.002 |
| | Sand-30 mm | 0.021 | 0.009 | 2.264 | 0.376 |
| | Control | −0.063 | 0.009 | −6.898 | < .001*** |
| Leca-30 mm | Sand-30 mm | −0.018 | 0.009 | −2.007 | 0.543 |
| | Control | −0.102 | 0.009 | −11.169 | < .001*** |
| Sand-30 mm | Control | −0.084 | 0.009 | −9.162 | < .001*** |

*p < .05, **p < .01, ***p < .001 (P-value adjusted for comparing a family of 9)

compacted from 1,030 ml to 816 ml, and the container filled to 60 mm from the top compacted from 780 ml to 615 ml. The WHC of the controls increased as the depth of medium decreased.

The results of the CV trials are presented in Fig 5 by combined drainage layer type. Post-hoc Tukey comparisons are presented in Table 4 (by combined drainage layer type), Table 5 (by drainage substrate only), and Table 6 (by drainage layer depth only). Compared to the control, the gravel-30 mm, grit-30 mm, and leca-30 mm drainage layer conditions did not have significantly different overall WHC. The grit-60 mm ($p < 0.01$), gravel-60 mm, leca-60 mm, sand-60 mm, and sand-30 mm conditions ($p < 0.001$) all had lower WHC than the control. There were significant differences between many of the drainage layer conditions, with the 30 mm grit condition having the highest WHC and 60 mm gravel the lowest. Analysing only according to the drainage substrate, conditions with grit did not have significantly different overall WHC, while leca ($p < 0.05$), gravel ($p < 0.01$), and sand ($p <$

**Table 2. Post-hoc Tukey comparisons of water-holding capacity of the entire container of CPB medium with different drainage layer substrates.**

| Drainage layer substrate | Comparison | Mean Difference | SE | t | p |
|---|---|---|---|---|---|
| Gravel | Grit | −0.009 | 0.007 | −1.248 | 0.723 |
| | Leca | 0.022 | 0.007 | 3.169 | 0.018* |
| | Sand | 0.010 | 0.007 | 1.417 | 0.619 |
| | Control | −0.084 | 0.008 | −9.914 | < .001*** |
| Grit | Leca | 0.031 | 0.007 | 4.417 | < .001*** |
| | Sand | 0.018 | 0.007 | 2.665 | 0.068 |
| | Control | −0.075 | 0.008 | −8.895 | < .001*** |
| Leca | Sand | −0.012 | 0.007 | −1.752 | 0.408 |
| | Control | −0.106 | 0.008 | −12.502 | < .001*** |
| Sand | Control | −0.094 | 0.008 | −11.071 | < .001*** |

*p < .05, **p < .01, ***p < .001 (P-value adjusted for comparing a family of 5)

**Table 3. Post-hoc Tukey comparisons of water-holding capacity of the entire container of CPB medium with different drainage layer depths.**

| Drainage layer depth | Comparison | Mean Difference | SE | t | p |
|---|---|---|---|---|---|
| 60 mm | 30 mm | −0.009 | 0.005 | −1.676 | 0.220 |
| | Control | −0.094 | 0.008 | −11.181 | < .001*** |
| 30 mm | Control | −0.085 | 0.008 | −10.121 | < .001*** |

*p < .05, **p < .01, ***p < .001 (P-value adjusted for comparing a family of 3)

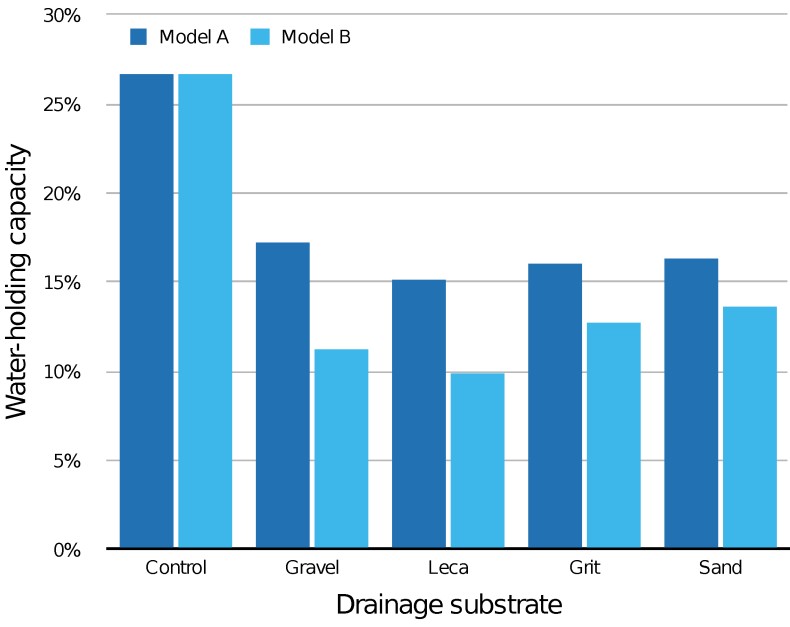

**Fig 3. Estimated mean water-holding capacity (WHC) of the CPB medium alone with different 30 mm drainage layers, according to models A and B.**

0.001) all had lower WHC than the control. Sand had the lowest WHC, and was significantly different ($p < 0.001$) from grit which had the highest WHC. Combining all substrates, 60 mm

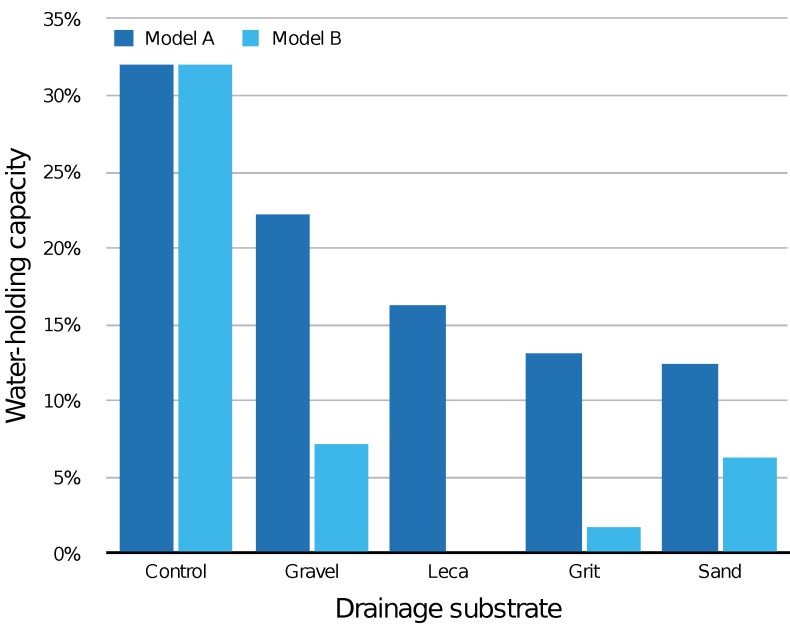

**Fig 4. Estimated mean water-holding capacity (WHC) of the CPB medium alone with different 60 mm drainage layers, according to models A and B.**

layers had lower overall WHC ($p < 0.001$). 30 mm layers were not significantly different from the control, but had significantly higher WHC than 60 mm layers ($p < 0.001$).

The modelled WHC of the medium itself according to drainage layer substrate is presented in Fig 6 (30 mm drainage layers) and Fig 7 (60 mm drainage layers), compared with the control containers with the same initial medium depth. With both drainage layer depths, all the modelled WHC values were lower than the control values, with the 60 mm drainage layers showing the greatest difference.

## John Innes medium

The JI mix showed the most compaction, with the effective volume of mix in a full pot decreasing from 1,290 ml to 925 ml. The control container filled to 30 mm from the top compacted from 1,030 ml to 800 ml, and the container filled to 60 mm from the top compacted from 780 ml to 590 ml. The WHC of the controls was lowest for the full pot and highest for the pot filled to 60 mm from the top.

The results of the JI trials are presented in Fig 8 by combined drainage layer type. Post-hoc Tukey comparisons are presented in Table 7 (by combined drainage layer type), Table 8 (by drainage substrate only), and Table 9 (by drainage layer depth only). Compared to the control, the leca-30 mm drainage layer condition had higher overall WHC ($p < 0.05$), the sand-60 mm drainage layer condition had lower overall WHC ($p < 0.01$), and all other conditions were not significantly different. Analysing only according to drainage substrate, sand had lower overall WHC ($p < 0.05$), and other substrates were not significantly different. Sand had the lowest overall WHC, and was significantly different from all other substrates, while leca had the highest WHC. Combining all substrates, neither 30 mm or 60 mm drainage layers had significantly different overall WHC from the control.

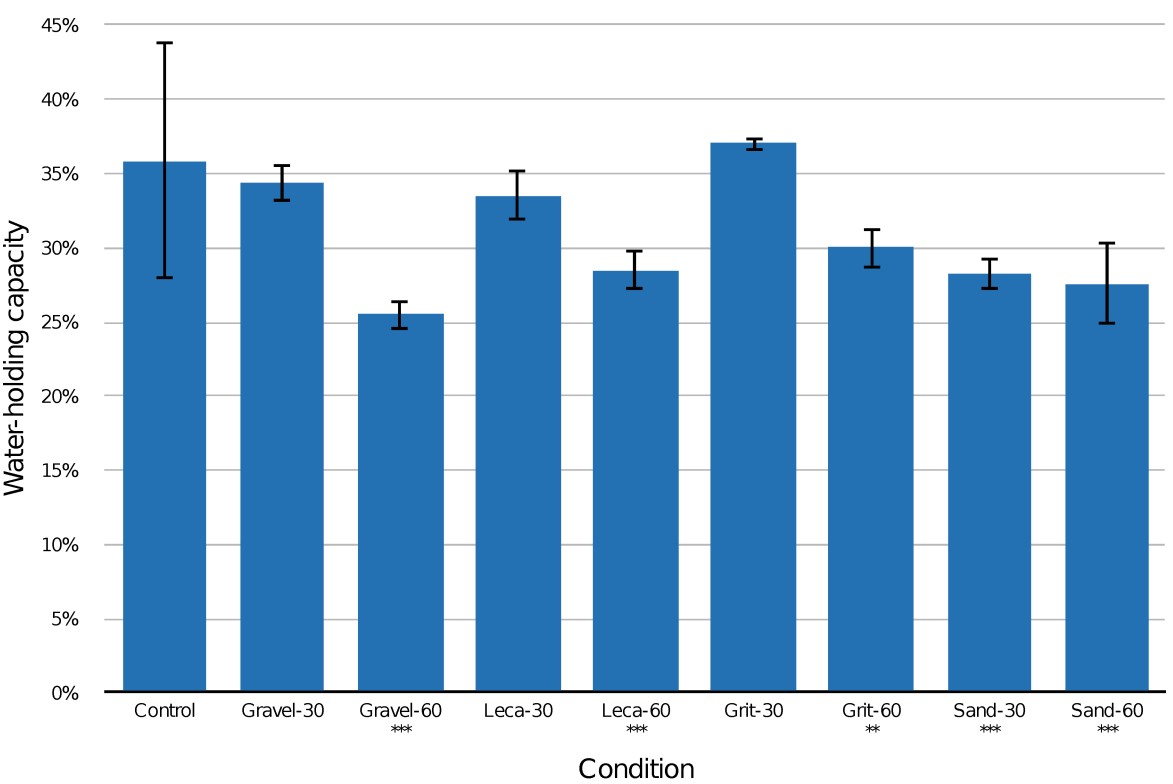

**Fig 5. Mean and standard deviation water-holding capacity (WHC) of the entire container of CV medium with different drainage layers.** Asterisks mark conditions which were significantly different from the control (**$p < 0.01$, ***$p < 0.001$).

The modelled WHC of the medium itself in the drainage layer conditions is presented in Fig 9 (30 mm drainage layers) and Fig 10 (60 mm drainage layers), compared with the control containers with the same initial medium depth. For 30 mm drainage layers, grit and sand had lower WHC than the control according to both models, while gravel and leca had modelled WHC values both above and below the control. For 60 mm drainage layers, sand had lower WHC than the control according to both models, while all other substrates had modelled WHC values both above and below the control.

## Discussion

The results show that adding a layer of coarse substrate below potting media will usually reduce water retention of the container overall, sometimes have no effect, and is very unlikely to increase water retention. This supports the recommendation of adding drainage layers to increase drainage, and contradicts previous theoretical predictions that such layers worsen drainage [12,13].

The addition of drainage layers caused a greater reduction in water retention when the potting media was coarser and more porous. This may be due to the similarity in pore size improving the capillary movement of water through the interface between different materials [18]. This possible explanation is also supported by the fact that the loam-based John Innes media had the greatest reduction in water retention when using a substrate of 1–2 mm sand, which has the smallest particle size and therefore a pore size closest to the JI medium itself.

**Table 4. Post-hoc Tukey comparisons of water-holding capacity of the entire container of CV medium with different drainage layer types.**

| Drainage layer type | Comparison | Mean Difference | SE | t | p |
|---|---|---|---|---|---|
| Control | Gravel-60 mm | 0.104 | 0.014 | 7.607 | < .001*** |
|  | Gravel-30 mm | 0.014 | 0.014 | 1.052 | 0.979 |
|  | Grit-60 mm | 0.058 | 0.014 | 4.291 | 0.002** |
|  | Grit-30 mm | −0.011 | 0.014 | −0.802 | 0.997 |
|  | Leca-60 mm | 0.073 | 0.014 | 5.371 | < .001*** |
|  | Leca-30 mm | 0.024 | 0.014 | 1.718 | 0.733 |
|  | Sand-60 mm | 0.082 | 0.014 | 6.043 | < .001*** |
|  | Sand-30 mm | 0.076 | 0.014 | 5.613 | < .001*** |
| Gravel-60 mm | Gravel-30 mm | −0.089 | 0.014 | −6.555 | < .001*** |
|  | Grit-60 mm | −0.045 | 0.014 | −3.316 | 0.035* |
|  | Grit-30 mm | −0.115 | 0.014 | −8.409 | < .001*** |
|  | Leca-60 mm | −0.030 | 0.014 | −2.236 | 0.393 |
|  | Leca-30 mm | −0.080 | 0.014 | −5.686 | < .001*** |
|  | Sand-60 mm | −0.021 | 0.014 | −1.565 | 0.821 |
|  | Sand-30 mm | −0.027 | 0.014 | −1.994 | 0.552 |
| Gravel-30 mm | Grit-60 mm | 0.044 | 0.014 | 3.239 | 0.044* |
|  | Grit-30 mm | −0.025 | 0.014 | −1.854 | 0.647 |
|  | Leca-60 mm | 0.059 | 0.014 | 4.319 | 0.001** |
|  | Leca-30 mm | 0.010 | 0.014 | 0.695 | 0.999 |
|  | Sand-60 mm | 0.068 | 0.014 | 4.991 | < .001*** |
|  | Sand-30 mm | 0.062 | 0.014 | 4.561 | < .001*** |
| Grit-60 mm | Grit-30 mm | −0.069 | 0.014 | −5.093 | < .001*** |
|  | Leca-60 mm | 0.015 | 0.014 | 1.080 | 0.976 |
|  | Leca-30 mm | −0.034 | 0.014 | −2.458 | 0.268 |
|  | Sand-60 mm | 0.024 | 0.014 | 1.752 | 0.713 |
|  | Sand-30 mm | 0.018 | 0.014 | 1.322 | 0.922 |
| Grit-30 mm | Leca-60 mm | 0.084 | 0.014 | 6.173 | < .001*** |
|  | Leca-30 mm | 0.035 | 0.014 | 2.499 | 0.248 |
|  | Sand-60 mm | 0.093 | 0.014 | 6.845 | < .001*** |
|  | Sand-30 mm | 0.087 | 0.014 | 6.415 | < .001*** |
| Leca-60 mm | Leca-30 mm | −0.049 | 0.014 | −3.509 | 0.020* |
|  | Sand-60 mm | 0.009 | 0.014 | 0.672 | 0.999 |
|  | Sand-30 mm | 0.003 | 0.014 | 0.242 | 1.000 |
| Leca-30 mm | Sand-60 mm | 0.058 | 0.014 | 4.163 | 0.002** |
|  | Sand-30 mm | 0.052 | 0.014 | 3.745 | 0.010** |
| Sand-60 mm | Sand-30 mm | −0.006 | 0.014 | −0.430 | 1.000 |

*p < .05, **p < .01, ***p < .001 (P-value adjusted for comparing a family of 9)

Thicker layers of drainage substrate tended to increase drainage more than thinner layers. This could be due to the perched water table of the drainage substrate itself making up a smaller proportion of the volume of the deeper layer. There was no apparent difference in drainage effects between porous and non-porous drainage substrates of the same particle size (leca and gravel). This is likely because leca is slow to absorb water, and holds onto water at high tensions [33]. Thus, the leca in the drainage layer would not absorb significant quantities of water in the short time the container is saturated before drainage, and would therefore behave similarly to non-porous gravel of the same particle size.

For a very coarse loamless medium such as the CPB mix, the results show that any drainage substrate of any depth can be used to good effect to increase drainage. For a finer loamless medium such as the CV mix, a 60 mm drainage layer is recommended, but any substrate is effective. For a loam-based medium such as John Innes, a 60 mm layer of coarse sand is recommended for the greatest increase in drainage.

**Table 5. Post-hoc Tukey comparisons of water-holding capacity of the entire container of CV medium with different drainage layer substrates.**

| Drainage layer substrate | Comparison | Mean Difference | SE | t | p |
|---|---|---|---|---|---|
| Control | Gravel | 0.059 | 0.016 | 3.607 | 0.005** |
| | Grit | 0.024 | 0.016 | 1.453 | 0.595 |
| | Leca | 0.050 | 0.017 | 3.025 | 0.027* |
| | Sand | 0.079 | 0.016 | 4.855 | < .001*** |
| Gravel | Grit | −0.035 | 0.013 | −2.637 | 0.073 |
| | Leca | −0.009 | 0.014 | −0.671 | 0.962 |
| | Sand | 0.020 | 0.013 | 1.529 | 0.547 |
| Grit | Leca | 0.026 | 0.014 | 1.932 | 0.309 |
| | Sand | 0.056 | 0.013 | 4.166 | < .001*** |
| Leca | Sand | 0.029 | 0.014 | 2.180 | 0.197 |

*p < .05, **p < .01, ***p < .001 (P-value adjusted for comparing a family of 5)

**Table 6. Post-hoc Tukey comparisons of water-holding capacity of the entire container of CV medium with different drainage layer depths.**

| Drainage layer depth | Comparison | Mean Difference | SE | t | p |
|---|---|---|---|---|---|
| Control | 60 mm | 0.079 | 0.014 | 5.879 | < .001*** |
| | 30 mm | 0.026 | 0.014 | 1.923 | 0.138 |
| 60 mm | 30 mm | −0.053 | 0.009 | −6.209 | < .001*** |

*p < .05, **p < .01, ***p < .001 (P-value adjusted for comparing a family of 3)

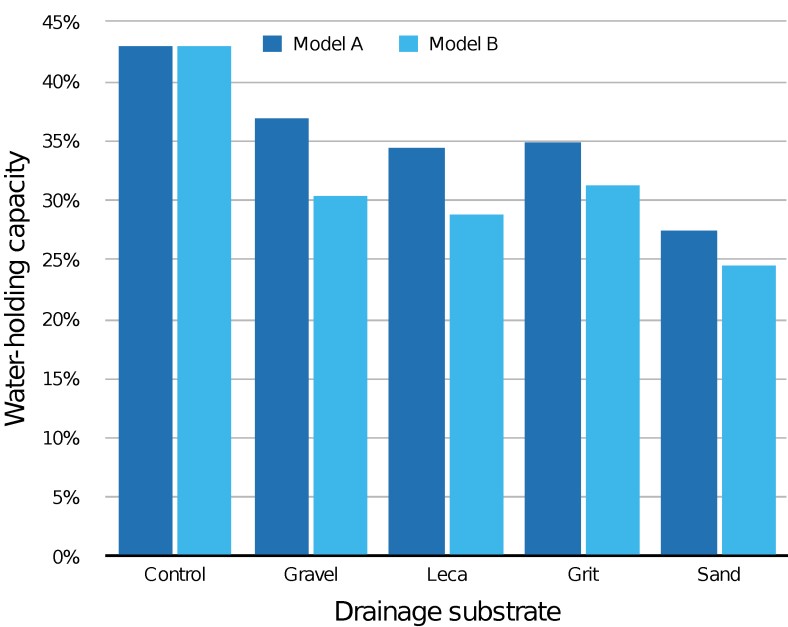

**Fig 6. Estimated mean water-holding capacity (WHC) of the CV medium alone with different 30 mm drainage layers, according to models A and B.**

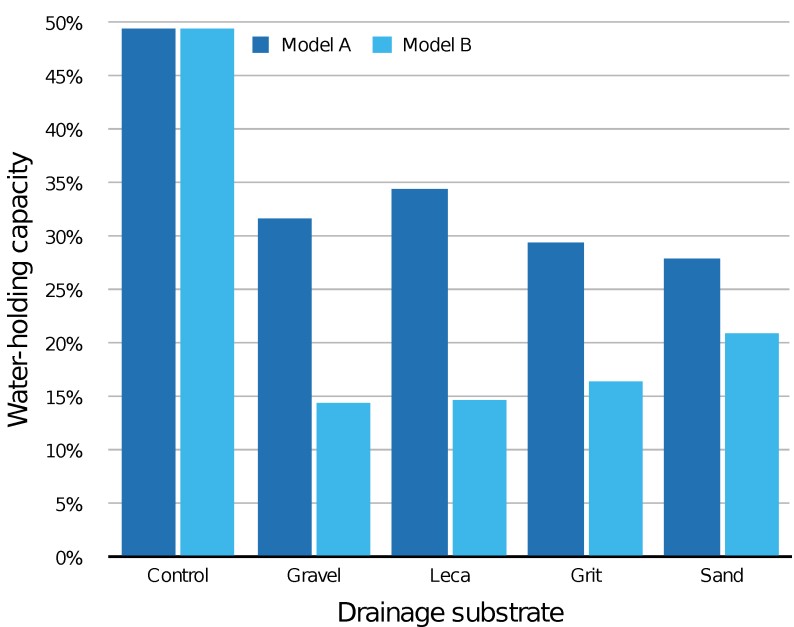

**Fig 7. Estimated mean water-holding capacity (WHC) of the CV medium alone with different 60 mm drainage layers, according to models A and B.**

The two different models for WHC of the potting medium alone were intended to provide the widest possible range of values, as a starting point for analysis. From visual examination (Fig 11), the drainage layers in containers with loamless potting media appeared to have approximately the same perched water table depth as when alone. Thus, for the CV and CPB media, the predictions of model A are likely to be most accurate for the WHC of the medium alone. For these media both models showed that all drainage layers reduced WHC, so the choice of model for analysis does not significantly change the conclusions.

In contrast, the drainage layers in containers with JI medium appeared to remain almost completely saturated after drainage (Fig 12). Thus, for JI medium the predictions of model B are likely to be most accurate for the WHC of the medium alone. The predictions of model B showed that all substrates decreased WHC, whereas model A showed a decrease for some substrates and an increase for others. Thus if model B is considered to be more accurate, it can be concluded that all drainage layers reduced WHC in the medium itself, whereas only some drainage layers reduced WHC in the container as a whole. More detailed study is required to confirm these predictions and validate the models.

The present study had a narrow focus, aiming to analyse the real-world behaviour of water moving through potting media in containers with drainage layers. The individual trials were controlled to ensure validity of the results for the values being measured. However there were no further laboratory investigations into the pore and grain size, quantitative water potential, air-entry value, or other characteristics of the media containers in isolation. These characteristics have a significant impact on the water behaviour of media [7]. Therefore a valuable direction for future research would be to replicate the experiment alongside further laboratory analysis of the media containers, to deepen understanding of why the results of the present study were found. Replicating the experiment using a range of different potting media, drainage substrates, and container shapes and sizes would allow quantitative analysis of how these factors affect the impact of drainage layers on water retention.

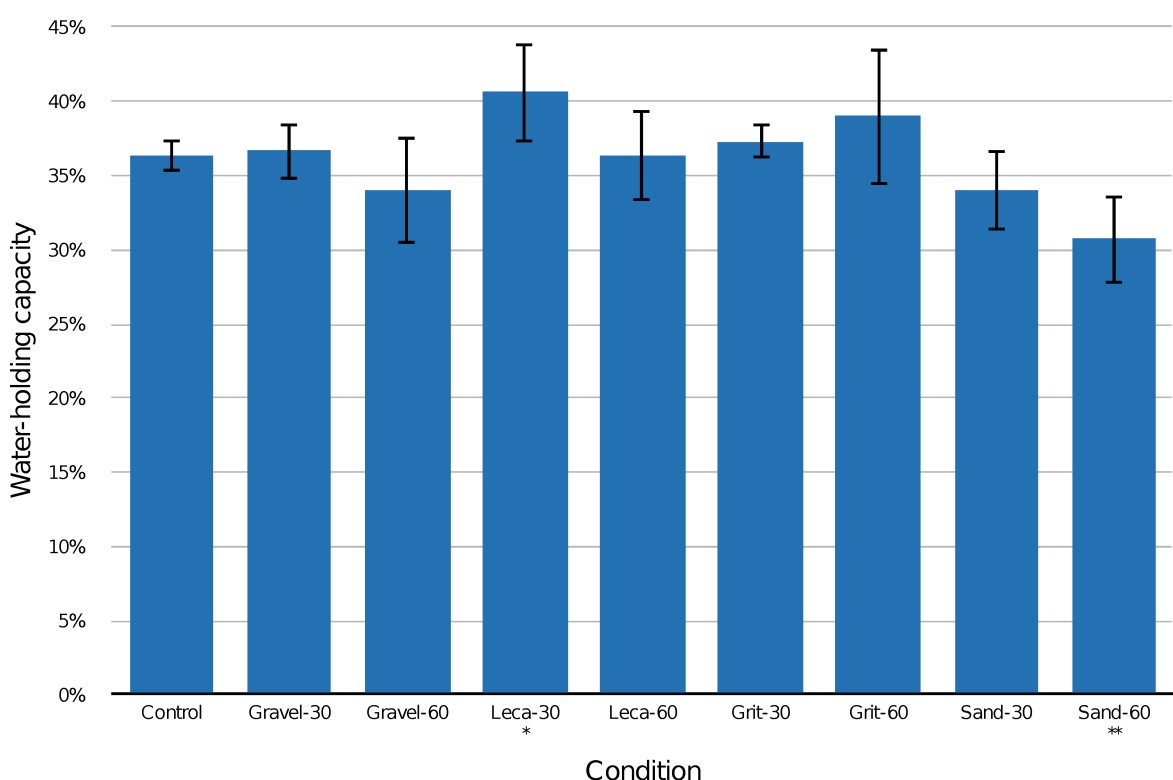

**Fig 8. Mean and standard deviation water-holding capacity (WHC) of the entire container of JI medium with different drainage layers.** Asterisks mark conditions which were significantly different from the control (*$p < 0.05$, **$p < 0.01$).

The choice to emulate real-world conditions also prevented laboratory controls being applied to the methods of filling, watering, and draining the test containers. Instead, simple practical methods such as weighing media and visual observation of the container contents and drainage process were employed to control the conditions as well as possible. The consistency between results within each experimental condition suggests that these controls were sufficient to produce valid results. However, further replications of the study will be valuable to confirm its validity.

In the present study, containers were fully saturated before being drained, and were assumed to have 100% of their pore spaces filled with water before drainage began. This allowed analysis of the drainage behaviour of the different media combinations in isolation from their wetting behaviour. However in practical use containers do not reach full saturation when they are watered from above, because water begins to drain out before the point of saturation [21], and because some air becomes entrapped in isolated pore spaces. Investigations of soil water hysteresis have shown that soils typically retain more water at a given suction level while drying than while wetting [34,35]. This is largely governed by the 'ink-bottle effect', in which large pores with small openings remain filled with water during drying. Additionally, many studies have shown dynamic non-equilibrium effects in which a given medium at a given suction level can have differing water content depending on whether the water is static, moving with a steady flow, or moving with a variable flow [36]. There are many possible causes for these effects and they are still poorly-understood, with predictions differing between research conducted in different fields [37]. Therefore further study is required

**Table 7. Post-hoc Tukey comparisons of water-holding capacity of the entire container of JI medium with different drainage layer types.**

| Drainage layer type | Comparison | Mean Difference | SE | t | p |
|---|---|---|---|---|---|
| Control | Gravel-60 mm | 0.021 | 0.013 | 1.619 | 0.792 |
| | Gravel-30 mm | −0.004 | 0.013 | −0.306 | 1.000 |
| | Grit-60 mm | −0.027 | 0.013 | −2.104 | 0.478 |
| | Grit-30 mm | −0.010 | 0.013 | −0.766 | 0.997 |
| | Leca-60 mm | −0.001 | 0.013 | −0.098 | 1.000 |
| | Leca-30 mm | −0.044 | 0.013 | −3.359 | 0.031* |
| | Sand-60 mm | 0.055 | 0.013 | 4.229 | 0.002** |
| | Sand-30 mm | 0.021 | 0.013 | 1.645 | 0.777 |
| Gravel-60 mm | Gravel-30 mm | −0.025 | 0.013 | −1.925 | 0.599 |
| | Grit-60 mm | −0.049 | 0.013 | −3.723 | 0.010* |
| | Grit-30 mm | −0.031 | 0.013 | −2.385 | 0.307 |
| | Leca-60 mm | −0.022 | 0.013 | −1.717 | 0.735 |
| | Leca-30 mm | −0.065 | 0.013 | −4.978 | < .001*** |
| | Sand-60 mm | 0.034 | 0.013 | 2.610 | 0.199 |
| | Sand-30 mm | $3.436 \times 10^{-4}$ | 0.013 | 0.026 | 1.000 |
| Gravel-30 mm | Grit-60 mm | −0.023 | 0.013 | −1.798 | 0.683 |
| | Grit-30 mm | −0.006 | 0.013 | −0.460 | 1.000 |
| | Leca-60 mm | 0.003 | 0.013 | 0.208 | 1.000 |
| | Leca-30 mm | −0.040 | 0.013 | −3.053 | 0.071 |
| | Sand-60 mm | 0.059 | 0.013 | 4.535 | < .001*** |
| | Sand-30 mm | 0.025 | 0.013 | 1.951 | 0.581 |
| Grit-60 mm | Grit-30 mm | 0.017 | 0.013 | 1.338 | 0.917 |
| | Leca-60 mm | 0.026 | 0.013 | 2.007 | 0.543 |
| | Leca-30 mm | −0.016 | 0.013 | −1.255 | 0.941 |
| | Sand-60 mm | 0.083 | 0.013 | 6.333 | < .001*** |
| | Sand-30 mm | 0.049 | 0.013 | 3.749 | 0.010** |
| Grit-30 mm | Leca-60 mm | 0.009 | 0.013 | 0.668 | 0.999 |
| | Leca-30 mm | −0.034 | 0.013 | −2.593 | 0.206 |
| | Sand-60 mm | 0.065 | 0.013 | 4.995 | < .001*** |
| | Sand-30 mm | 0.031 | 0.013 | 2.411 | 0.292 |
| Leca-60 mm | Leca-30 mm | −0.043 | 0.013 | −3.261 | 0.041* |
| | Sand-60 mm | 0.056 | 0.013 | 4.327 | 0.001** |
| | Sand-30 mm | 0.023 | 0.013 | 1.743 | 0.718 |
| Leca-30 mm | Sand-60 mm | 0.099 | 0.013 | 7.588 | < .001*** |
| | Sand-30 mm | 0.065 | 0.013 | 5.004 | < .001*** |
| Sand-60 mm | Sand-30 mm | −0.034 | 0.013 | −2.584 | 0.210 |

*p < .05, **p < .01, ***p < .001 (P-value adjusted for comparing a family of 9)

to investigate the effects of hysteresis and dynamic non-equilibrium on the water movement through containers with drainage layers during irrigation.

The present study did not test the response of living plants when grown in containers with drainage layers. Although the results can be applied effectively to decrease water retention of a given container of media, they cannot be used to predict whether a given plant will benefit from this intervention. Recent studies have shown that stratified containers with a more porous medium in the lower half can improve crop growth, quality, and yield in experimental conditions [29]. Controlled growing trials are required to study the growth of plants in containers with shallow drainage layers, of the type considered in this study and commonly employed by horticulturists and home growers. These trials could investigate a range of factors such as crop yield and quality, irrigation requirements, shoot and root behaviour, and the differing responses of different plant species.

**Table 8. Post-hoc Tukey comparisons of water-holding capacity of the entire container of JI medium with different drainage layer substrates.**

| Drainage layer substrate | Comparison | Mean Difference | SE | t | p |
|---|---|---|---|---|---|
| Control | Gravel | 0.009 | 0.012 | 0.686 | 0.959 |
| | Grit | −0.019 | 0.012 | −1.499 | 0.566 |
| | Leca | −0.023 | 0.012 | −1.806 | 0.377 |
| | Sand | 0.038 | 0.012 | 3.069 | 0.023* |
| Gravel | Grit | −0.027 | 0.010 | −2.676 | 0.066 |
| | Leca | −0.031 | 0.010 | −3.052 | 0.025* |
| | Sand | 0.030 | 0.010 | 2.919 | 0.035* |
| Grit | Leca | −0.004 | 0.010 | −0.375 | 0.996 |
| | Sand | 0.057 | 0.010 | 5.595 | < .001*** |
| Leca | Sand | 0.061 | 0.010 | 5.970 | < .001*** |

*p < .05, **p < .01, ***p < .001 (P-value adjusted for comparing a family of 5)

**Table 9. Post-hoc Tukey comparisons of water-holding capacity of the entire container of JI medium with different drainage layer depths.**

| Drainage layer depth | Comparison | Mean Difference | SE | t | p |
|---|---|---|---|---|---|
| Control | 60 mm | 0.012 | 0.014 | 0.879 | 0.655 |
| | 30 mm | −0.009 | 0.014 | −0.671 | 0.781 |
| 60 mm | 30 mm | −0.021 | 0.009 | −2.452 | 0.042* |

*p < .05, **p < .01, ***p < .001 (P-value adjusted for comparing a family of 3)

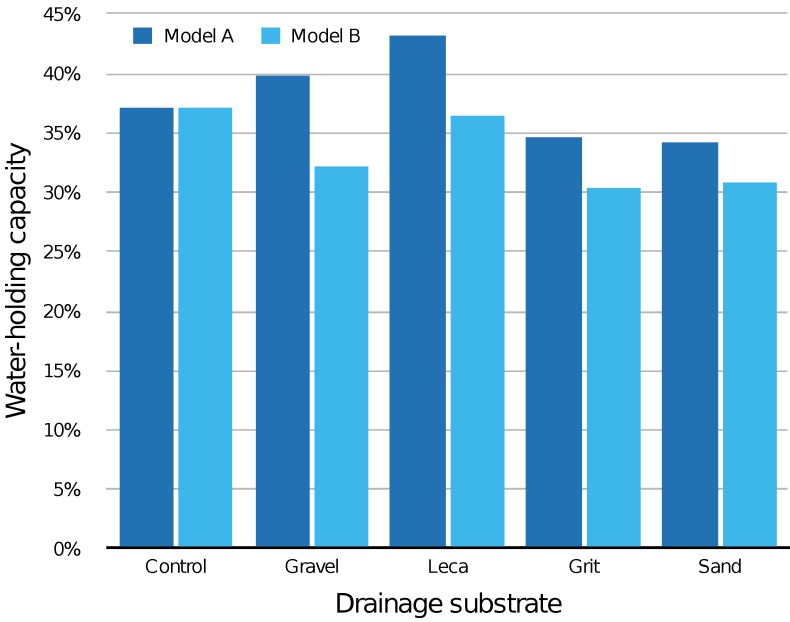

**Fig 9. Estimated mean water-holding capacity (WHC) of the JI medium alone with different 30 mm drainage layers, according to models A and B.**

## Conclusions

Despite being frequently discussed by horticulturists and home gardeners, there has been little applied research into the effects of drainage layers on the water-holding capacity of container

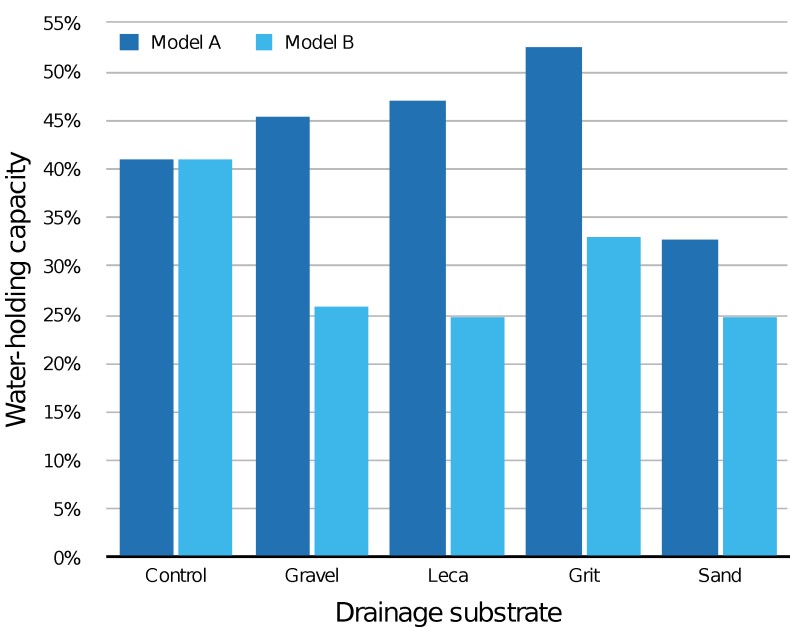

**Fig 10. Estimated mean water-holding capacity (WHC) of the JI medium alone with different 60 mm drainage layers, according to models A and B.**

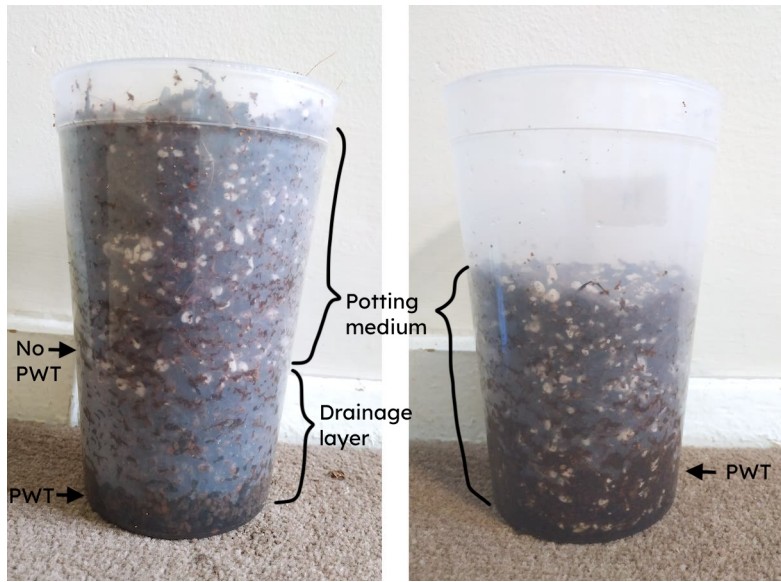

**Fig 11. Comparison of CPB medium with 60 mm layer of grit (left) to CPB medium alone to a depth of 60 mm below the top (right).** In the left image, the perched water table (PWT) in the medium is completely eliminated, and the low perched water table in the grit is visible.

media. Related research in different fields has led to conflicting predictions about whether drainage layers will increase, or conversely decrease, the amount of water stored in the soil above. The object of this study was to experimentally determine how the addition of a layer of coarse substrate at the bottom of a container would affect the water-holding capacity of

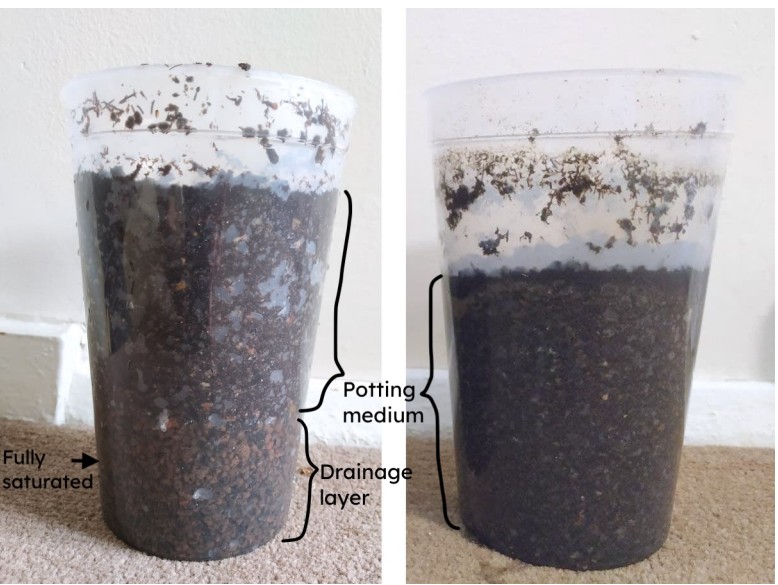

**Fig 12. Comparison of JI medium with 60 mm layer of grit (left) to JI medium alone to a depth of 60 mm below the top (right).** In the left image, the grit layer appears to be completely saturated after drainage.

the potting medium above. Three experiments were conducted using three different common potting media. Each experiment varied the depth and substrate of the drainage layer, and measured the amount of water retained in the container. Two simple models were used to estimate the water retention in the medium alone by subtracting the water retained in the drainage layer itself. The potting media, drainage substrates, containers, and watering methods were chosen to emulate the materials and methods typically available to home gardeners, to ensure the results would be valid for practical application.

Drainage layers were almost universally found to either decrease or have no effect on the water-holding capacity of containers. Thicker drainage layers decreased water retention more than thinner layers, but the most effective drainage substrate to use was dependent on the specific potting medium. Coarser organic potting media showed the greatest reduction in water-holding capacity with the addition of drainage layers, and fine loam-based media the least. For loamless organic media, both models agreed that all drainage layers reduced the water retained in the medium alone. For loam-based media the models differed in their estimates, but the most plausible model based on visual examination showed that all drainage layers reduced water retention of the medium alone.

The present study was conducted using method and material limitations similar to those of home gardeners. This was intended to ensure the results would be valid for practical application in their intended context. The result can be taken to show that warnings about the raised perched water table and increased water retention should be considered inaccurate. Instead, drainage layers can be considered a potentially useful tool for growers and home gardeners to reduce the water retention of a given potting medium. A 60 mm later of coarse sand was the most universally-effective drainage layer with all potting media tested and therefore the best single recommendation for growers wishing to increase drainage. Due to the focus of the research on practical application, the pore and grain size, quantitative water potential, air-entry value, and other characteristics of the potting media and drainage substrates were not measured. Thus the results cannot be modelled in relation to these metrics, and gaining

improved understanding of why the present results were obtained would be a valuable focus for future research.

## Supporting information

**S1 Appendix. ANOVA tables.**
(PDF)

**S1 Data. Raw data from measurements of CPB media trials.**
(CSV)

**S2 Data. Raw data from measurements of CV media trials.**
(CSV)

**S3 Data. Raw data from measurements of JI media trials.**
(CSV)

**S4 Data. Raw data from measurements of drainage layer trials without potting media.**
(CSV)

## Acknowledgments

The research received no external support.

## Author contributions

**Conceptualization:** Avery Rowe.

**Data curation:** Avery Rowe.

**Formal analysis:** Avery Rowe.

**Investigation:** Avery Rowe.

**Methodology:** Avery Rowe.

**Project administration:** Avery Rowe.

**Resources:** Avery Rowe.

**Software:** Avery Rowe.

**Supervision:** Avery Rowe.

**Validation:** Avery Rowe.

**Visualization:** Avery Rowe.

**Writing – original draft:** Avery Rowe.

**Writing – review & editing:** Avery Rowe.

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
