## [Decision Letter · Decision Letter 0]

2 Oct 2024

PONE-D-24-32597Effect of drainage layers on water retention of potting media in containersPLOS ONE

Dear Dr. Rowe,

Thank you for submitting your manuscript to PLOS ONE. After careful consideration, we feel that it has merit but does not fully meet PLOS ONE’s publication criteria as it currently stands. Therefore, we invite you to submit a revised version of the manuscript that addresses the points raised during the review process.

We look forward to receiving your revised manuscript.

Kind regards,

Somayeh Soltani-Gerdefaramarzi, Ph. D.

Academic Editor

PLOS ONE

Journal Requirements: When submitting your revision, we need you to address these additional requirements. 1. Please ensure that your manuscript meets PLOS ONE's style requirements, including those for file naming. The PLOS ONE style templates can be found at https://journals.plos.org/plosone/s/file?id=wjVg/PLOSOne_formatting_sample_main_body.pdf and https://journals.plos.org/plosone/s/file?id=ba62/PLOSOne_formatting_sample_title_authors_affiliations.pdf 2. Please include captions for your Supporting Information files at the end of your manuscript, and update any in-text citations to match accordingly. Please see our Supporting Information guidelines for more information: http://journals.plos.org/plosone/s/supporting-information.

Reviewers' comments:

Reviewer's Responses to Questions

**Comments to the Author**

1. Is the manuscript technically sound, and do the data support the conclusions?

Reviewer #1: Partly

Reviewer #2: Partly

2. Has the statistical analysis been performed appropriately and rigorously? 

Reviewer #1: No

Reviewer #2: Yes

3. Have the authors made all data underlying the findings in their manuscript fully available?

Reviewer #1: No

Reviewer #2: Yes

4. Is the manuscript presented in an intelligible fashion and written in standard English?

Reviewer #1: Yes

Reviewer #2: No

5. Review Comments to the Author

Reviewer #1: This paper presents an interesting work on exploring the best settings for the drainage layer down under the plant pot in consideration of the perched water effect or capillary barrier effect. I recognised the significance of this work's contribution from a very gardening and soil hydrological perspective. I am very keen to assist the author in publishing this work. Nonetheless, in terms of soil physics, the depth of comprehension and discussion is still shallow. To assist the authors in publishing their work at the high quality that PLOS One truly needs, I decided to reach out to a major revision to see if the author can substantially improve their research presentation in this manuscript by following the specific comment below. Also, the author is most welcome to rebut in an acceptable manner if there are any disagreements or differences in understanding saturated and unsaturated soil physics.

1. In order to deepen the knowledge basis and discussion in this paper, you may want to have a careful review of the ink-bottle effect and capillary barrier or clay liner, just like what you described for perched water above any possible aquitard or aquiclude, although you just reassign such an engineering-scale geological condition into the scenario in a plant container or a pot.

2. In addition, you may want to expand your understanding of unsaturated and saturated soil further from a soil hydrological point of view with assistance from a few good references.

Yan, G.; Li, Z.; Galindo Torres, S.A.; Scheuermann, A.; Li, L. Transient Two-Phase Flow in Porous Media: A Literature Review and Engineering Application in Geotechnics. Geotechnics 2022, 2, 32-90. https://doi.org/10.3390/geotechnics2010003

Yan G, Bore T, Schlaeger S, Scheuermann A, Li L. Investigating scale effects in soil water retention curve via spatial time domain reflectometry. Journal of Hydrology. 2022 Sep 1;612:128238.

3. Even though you have presented some data to manifest the research findings you have in the abstract, you have not quantitatively analysed the grain size distributions of coarse, fine, and loamy layers with different settings in drainage layers. It would be much better if you could provide more sieving analysis for different configurations between the soil substrate at the top and the gravel-sand mixing drainage layer at the bottom. Have you checked the packing conditions in your plat pot (e.g., soil density, porosity, void ratio, etc.)? Different packing conditions will lead to different findings, and they ought to be carefully controlled and listed in your experimental settings for statistical analysis.

4. In the methodological section, you may want to insert more information about the measuring errors and uncertainties that you encountered during the tests and write up a good self-defence to strengthen the rigour and prudence of your experimental methods. Besides, you forgot to assign codes and numbers for all equations you left in this section, and neither did you for notations of all mathematical symbols. Also, experimental demos and visual illustrations are lacking in this method section.

5. Regarding the analysis of mean (ANOM) and analysis of variance (ANOVA), you can provide a table to illustrate how it was statistically calculated with all settings in such a design of experiment. Corresponding statistical equations can also be inserted into this section to enrich your mathematical complexity, but please do not forget to number them.

6. If possible, you can give more discussion on air-entry value, capillary infiltration breakthrough water pressure and its corresponding soil moisture storage, porosity (packing conditions), etc. Those will help you professionalise your discussion in addition to the relatively superficial understanding of the experimental data.

7. You can highlight what you intend to demonstrate with a few red boxes in the dashline in Figures 10 and 11.

8. The conclusions are all lumped into a single paragraph in a relatively poor writing structure. The conclusion ought to be in this sequence: Paragraph 1: (i) Concise summary of your research gaps identified through literature review and research objectives; (ii) Briefing your methodology with some very critical experimental settings; Paragraph 2: (iii) Listing your conclusions or research findings point-by-point; Paragraph 3: (iv) Mention the limitations and reflection of your work; (v) Highlight the significance of your research and engineering applications.

Reviewer #2: Dear Editor and Author

I hope this message finds you well. I would like to express my appreciation for the article, which I believe holds significant practical value. However, I have identified several points that should be addressed to enhance the transparency of the content. These are outlined below:

1. In line 130, it is noted that fiberglass mesh is used exclusively for the containers containing the sand drainage layer. Given that the size of the drainage hole at the end of the container is 10 mm, which exceeds the aggregate size of the substrate in the drainage layer, could you clarify this choice?

2. I recommend including a schematic diagram of the container to improve understanding.

3. It is unclear why the soil was not covered during the drainage period and why the duration of drainage was limited to only 3 hours. Typically, in a control pot, water drainage lasts longer. Please provide clarification on this matter.

4. More detailed information on how the containers were filled would be beneficial. Specifically, how was layering of the soil prevented during filling?

5. In line 154, it is stated that moisture levels in the media within containers are not uniform; however, line 139 mentions that the soil was air-dried to achieve uniform moisture. It would be helpful to explain this apparent contradiction for clarity.

6. In laboratory settings, it is common practice to saturate soil by introducing water from the bottom to allow trapped air to escape from the top. Was this method employed in your experiment?

7. How did you ensure that the pots were not blocked, particularly in the control pots that lacked a drainage layer?

8. At the beginning of the Materials and Methods section, media were numbered; however, there appears to be incomplete numbering.

9. In line 129 "The base had a slight domed area raised by 4mm relative to the edges". Why was this container chosen?

Lastly, I noticed several typographical and spelling errors throughout the article, many of which could be easily corrected using text editing software.

Thank you for considering these suggestions. I believe addressing these points will significantly enhance the quality and clarity of your work. Wishing you continued success.

Best regards,

6. PLOS authors have the option to publish the peer review history of their article (what does this mean?). If published, this will include your full peer review and any attached files.

Reviewer #1: No

Reviewer #2: No

---

## [Author Response · Author response to Decision Letter 1]

20 Dec 2024

Reviewer #1

1. I have added a consideration of soil water hysteresis and the ink-bottle effect (lines 302-313). Capillary barriers are discussed in the introduction (lines 27-36).

2. I have included a citation in the discussion of soil water hysteresis and the ink-bottle effect (line 309).

3. Unfortunately these investigations were not conducted at the time of the experiment, and the original media are no longer available to examine. I believe that the research still has practical value, particularly for applied horticulturists who may need to make decisions about containers and growing media without being able to conduct laboratory investigations of their own. I have added an acknowledgement of this limitation to the study in the discussion (lines 314-328).

4. I have specified the precision of the measurements in the method (lines 134 and 143). I have added definitions for the terms in the equations (lines 159-160, 165-166, 170-171), and numbered the equations separately from the text (equations 1-4). Unfortunately no other photographs were taken of the experimental setup, but I have added a schematic diagram (figure 1).

5. I have added full tables of the post-hoc comparisons to the text (Tables 1-9), and the ANOVA calculations as a supplementary file (S1 Appendix).

6. Unfortunately these investigations were not conducted at the time of the experiment, and the original media and apparatus are no longer available to examine. I believe that the research still has value as it presents a novel and significant result with practical relevance, and also invites further study to deepen understanding of the mechanisms involved. I have added an acknowledgement of this limitation to the study in the discussion (lines 314-328).

7. I have added annotations to figures 11 and 12.

8. I have restructured and expanded the conclusions section (lines 330-367).

Reviewer #2

1. Although the individual particle sizes were smaller than the drainage hole, the friction of the aggregate pieces together was sufficient to prevent the other substrates escaping from the drainage hole without the addition of the fibreglass mesh. I have explained this in the text (lines 119-120).

2. I have added a schematic diagram of the experimental procedure (figure 1).

3. Drainage took only a short time due to the small size of the containers and the potting media used (which have larger particle sizes than field loams). The final measurements were taken when water was no longer visibly draining from the container. I have explained this in the text (lines 140-142). The containers were left uncovered to match the conditions of a real-world growing container being watered. I have explained this in the text (lines 138-140). Due to the short drainage period, the effect of evaporation is likely to be negligible.

4. The containers were filled by hand. This was again chosen to represent the real-world application of drainage layers in practical gardening. The clear containers allowed observation of the potting media and no evidence of layering was observed during filling, watering, or drainage. I have added this to the text (lines 142-143). I have also added a consideration of these practical limitations to the discussion (lines 323-328).

5. The different potting media were not dried to the same moisture level as each other (the three media were treated as independent experiments and not compared to each other), but every sample of a given medium was dried to the same moisture level (for validity between the trials of each experiment). I have clarified this in the text (lines 129-132).

6. The water was poured from above as it typically would be during real-world watering of container plants (lines 136-137). I have added a consideration of the watering method to the discussion (lines 302-313).

7. No special procedure was employed, but the containers were monitored during drainage and no blockages or inconsistencies were observed. I have mentioned this in the text (lines 142-143).

8. "No. 1" is the name of the specific formulation of John Innes compost, rather than numbering to identify the media in the text. I have added quotation marks to make this clearer in the text (line 98).

9. This was chosen to represent the typical shape of a nursery container, which generally have a raised centre and a lowered border or feet to allow water to escape when placed on a flat surface. I have mentioned this in the text (lines 115-117).

---

## [Decision Letter · Decision Letter 1]

2 Jan 2025

PONE-D-24-32597R1Effect of drainage layers on water retention of potting media in containersPLOS ONE

Dear Dr. Rowe,

Thank you for submitting your manuscript to PLOS ONE. After careful consideration, we feel that it has merit but does not fully meet PLOS ONE’s publication criteria as it currently stands. Therefore, we invite you to submit a revised version of the manuscript that addresses the points raised during the review process.

We look forward to receiving your revised manuscript.

Kind regards,

Somayeh Soltani-Gerdefaramarzi, Ph. D.

Academic Editor

PLOS ONE

Journal Requirements:

Reviewers' comments:

Reviewer's Responses to Questions

**Comments to the Author**

1. If the authors have adequately addressed your comments raised in a previous round of review and you feel that this manuscript is now acceptable for publication, you may indicate that here to bypass the “Comments to the Author” section, enter your conflict of interest statement in the “Confidential to Editor” section, and submit your "Accept" recommendation.

Reviewer #1: All comments have been addressed

2. Is the manuscript technically sound, and do the data support the conclusions?

Reviewer #1: Yes

3. Has the statistical analysis been performed appropriately and rigorously? 

Reviewer #1: Yes

4. Have the authors made all data underlying the findings in their manuscript fully available?

Reviewer #1: Yes

5. Is the manuscript presented in an intelligible fashion and written in standard English?

Reviewer #1: Yes

6. Review Comments to the Author

Reviewer #1: I do believe that the author has made the best effort to improve the academic rigour and quality of the manuscript from the last round of revisions. The author has not only provided more statistical analysis in addition to the previous bar charts but also significantly deepened his/her understanding of unsaturated soil science and capillarity in terms of soil suction, moisture and ink-bottle effects. Therefore, I recommend consideration of publication as it is decent work for in-door planting rather than unsaturated soil mechanics and vadose zone seepage science.

However, prior to formal acceptance, I would like to invite the author to reflect on their experimental limitations and then write up some good proposals for future research at the end of the discussion section. Regarding this limitation and prospect, other works deserve to be reviewed and cited herein.

Diamantopoulos E, Durner W (2012) Dynamic nonequilibrium of water flow in porous media: a review. Vadose Zone J 11(3):vzj2011.0197

Yan, G., Bore, T., Schlaeger, S., Scheuermann, A., & Li, L. (2024). Dynamic effects in soil water retention curves: An experimental exploration by full-scale soil column tests using spatial time-domain reflectometry and tensiometers. Acta Geotechnica, 1-27.

After all, dynamic nonequilibrium soil suction also matters with local heterogeneity, e.g., ink-bottle effects, further inducing concerns of the so-called “snap-off” and “Haine jump” effects at the local microscale. With such a self-reflection section, you can defend yourself by mentioning those limits in your current work and demonstrate your in-depth understanding of unsaturated soil knowledge that will be used to orient your academic pursuits in the future.

7. PLOS authors have the option to publish the peer review history of their article (what does this mean?). If published, this will include your full peer review and any attached files.

Reviewer #1: No

---

## [Author Response · Author response to Decision Letter 2]

7 Jan 2025

Dear Reviewer,

Thank you again for your valuable feedback on my submission. I attach a revised manuscript and a tracked changes manuscript.

I have restructured and expanded the discussion. It now includes further references, and more detailed proposals for future research.

I hope these improvements meet with your approval, and I am grateful for your time and advice.

Yours sincerely,

Avery Rowe

---

## [Decision Letter · Decision Letter 2]

21 Jan 2025

Effect of drainage layers on water retention of potting media in containers

PONE-D-24-32597R2

Dear Dr. Rowe,

We’re pleased to inform you that your manuscript has been judged scientifically suitable for publication and will be formally accepted for publication once it meets all outstanding technical requirements.

Kind regards,

Somayeh Soltani-Gerdefaramarzi, Ph. D.

Academic Editor

PLOS ONE

Additional Editor Comments (optional):

Reviewers' comments:

Reviewer's Responses to Questions

**Comments to the Author**

1. If the authors have adequately addressed your comments raised in a previous round of review and you feel that this manuscript is now acceptable for publication, you may indicate that here to bypass the “Comments to the Author” section, enter your conflict of interest statement in the “Confidential to Editor” section, and submit your "Accept" recommendation.

Reviewer #1: All comments have been addressed

2. Is the manuscript technically sound, and do the data support the conclusions?

Reviewer #1: Yes

3. Has the statistical analysis been performed appropriately and rigorously? 

Reviewer #1: Yes

4. Have the authors made all data underlying the findings in their manuscript fully available?

Reviewer #1: Yes

5. Is the manuscript presented in an intelligible fashion and written in standard English?

Reviewer #1: Yes

6. Review Comments to the Author

Reviewer #1: The author has fully addressed the comment I left before. This manuscript can be accepted as is after two rounds of revisions.

7. PLOS authors have the option to publish the peer review history of their article (what does this mean?). If published, this will include your full peer review and any attached files.

Reviewer #1: No

---

## [Editor Report · Acceptance letter]

PONE-D-24-32597R2

PLOS ONE

Dear Dr. Rowe,

I'm pleased to inform you that your manuscript has been deemed suitable for publication in PLOS ONE. Congratulations! Your manuscript is now being handed over to our production team.

Kind regards,

on behalf of

Dr. Somayeh Soltani-Gerdefaramarzi

Academic Editor

PLOS ONE